# Embedding E-Learning in Accounting Modules: The Educators' Perspective

**Konrad Grabinski** [1], **Marcin Kedzior** [1,*], **Joanna Krasodomska** [1] **and Agnieszka Herdan** [2]

1   Department of Financial Accounting, Cracow University of Economics, Rakowicka, 27, 31-510 Krakow, Poland; kg@uek.krakow.pl (K.G.); joanna.krasodomska@uek.krakow.pl (J.K.)
2   Accounting and Finance Department, University of Greenwich, London SE10 9LS, UK; a.herdan@gre.ac.uk
*   Correspondence: kedziorm@uek.krakow.pl; Tel.: +48-12-293-56-94

**Abstract:** The aim of the paper is to investigate the benefits and drawbacks resulting from the implementation of e-learning in accounting modules among educators. The primary source of data was a questionnaire conducted among 79 accounting lecturers, employed by the leading Polish economic universities. The results of the survey have shown that e-learning is not widely used by accounting academics in Poland. The most important benefits of the e-courses included the enhancement of efficiency and flexibility of the teaching process. The most serious difficulties were an extensive amount of work associated with designing and updating course materials and technical problems. The effectiveness of e-learning techniques in teaching accounting subjects is determined by the easiness of e-learning delivery, more regular learning process, greater development of students' social competences during e-learning classes and a more effective process of verification of students' progress, in comparison with traditional classes. Furthermore, the study provides evidence that lecturers, who decided to use e-learning, perceive this way of teaching as more efficient, and at the same time more demanding, in comparison to traditional classes. The paper contributes to the understanding of the use of e-learning in accounting education and offers findings that might be useful for both policymakers and practitioners.

**Keywords:** e-learning; accounting; education; educators

## 1. Introduction

The Internet, social media and being online constitute an integral part of everyday life. There is a common expectation that these new sources of communication will be used in the process of education. Universities are looking for ways to use the opportunities provided by this new approach and incorporate it effectively, making this way of learning more engaging and efficient and enhancing the higher education system. E-learning means adopting electronic educational technology in learning and teaching. It can be implemented in various shapes and forms. It can be performed either as blended learning (where only part of the course is offered online), or as entire courses delivered online. It comprises webinars, lectures/videos on-demand, multi-media components (3D presentations, animations, hypertext, hypermedia); various other online activities [1]. The use of technology transformed course delivery to be partly or fully independent of time and place [1,2]. The European Commission [3] emphasizes that the use of new multimedia technologies and the Internet increases learning quality. It allows easier and wider access to educational facilities and provides opportunities for distant exchanges and collaboration.

In our study, we focus on accounting education. The accounting discipline is perceived as very practical but also rather difficult. It requires methodical work and systematic studies. So, moving from the traditional delivery in the classroom to online learning will require thinking outside the box.

There is a very limited amount of research in the area of e-learning in accounting education. A lot of Polish educators can see the changing trends in higher education and growing needs to adopt new methods. However, many institutions still prefer the traditional face-to-face approach. There is a lot of anxiety with online teaching, as to the amount of time it requires, and the training it needs. Polish academics are very often uncertain as to how efficient the process will be and whether the benefits will be greater than the cost involved. This paper will look at the attitudes of Polish accounting educators to this relatively new teaching method.

The paper aims to investigate the benefits and drawbacks, resulting from the implementation of e-learning in accounting modules among educators. To achieve this goal, the literature review has been undertaken and a survey has been developed. This survey has been conducted among accounting educators employed by the leading Polish Economic Universities, such as the Cracow University of Economics, Wroclaw University of Economics, Poznan University of Economics and Business, University of Lodz, Kozminski University, the University of Economics in Katowice, University of Szczecin, University of Gdansk, Nicholas Copernicus University in Torun, and Warsaw School of Economics.

Basically, our survey has exploratory character and is intended to investigate how the advantages and disadvantages of the e-learning in accounting modules are perceived by lecturers. We want to compare and confront visions and ideas concerning e-learning of two very different groups and profiles of lecturers. Finally, we intend to identify the reasons, for which academics are not inclined to engage in e-learning education. We conjecture that among academics in Poland, and probably in other Central and Eastern Europe countries (CEE), there is an unjustified misperception about the drawbacks of e-learning, which we are trying to reveal.

Main findings suggest that within the group of academics, who decided to use e-learning in accounting modules, the perceived most important advantages are: the easiness of e-learning delivery in comparison with traditional methods, more regular learning process during, a greater degree of development of students' social competences, and finally a more effective process of verification of students' progress. Most notably, these advantages were not indicated by the other group—lecturers, who have not yet used e-learning. The results also show that e-learning techniques are undertaken by academics who are dedicated to education and are not avoiding the effort to deliver a high quality of teaching. The most important determinant motivating lecturers to invest in e-learning is better communication with students, understood as more frequent, and more direct. The most significant impediments of e-learning applications are technical problems and a sense of excessive mechanization. The results also suggest that academics using only traditional methods of teaching very often misperceive the real benefits and difficulties of e-learning.

Our study contributes to the literature on accounting education, as it provides new insights into the use of e-learning as an education supporting tool from the educators' perspective. To the best of our knowledge, most of the studies regarding factors that affect satisfaction with e-learning courses analyzed this problem from the student's point of view. The results are also relevant for practice. They might be useful for university authorities, which may introduce procedures facilitating such classes and, therefore, contribute to greater acceptability of them among lecturers. Lecturers may be more aware of the real benefits and disadvantages of conducting e-learning classes. The research findings can also be useful for other scientists and policymakers. We also believe that the results of our analysis might be extended to other Eastern European countries, due to cultural similarities existing in this geographical area. The effectiveness of e-learning courses is also conditioned by cultural factors [4,5]. In Eastern European countries, there are also similar systems, problems and challenges in the field of academic education, and the problem with access to high-speed internet [6].

The paper is organized as follows. The next section presents a review of literature on e-learning benefits and challenges, as well as the concept of e-learning effectiveness. Subsequently, the educators' approach to e-learning is discussed. This is followed by the empirical section, which provides information on the purpose and methods of research and offers a discussion of the study findings.

The last section presents the conclusions and limitations of the study, together with indications for possible future research.

## 2. Benefits, Challenges, and Effectiveness of E-Learning

There are some possible benefits of e-learning for students, educators, and higher education institutions. However, each of these participants in the educational process will face many challenges. For students, the most difficult ones include good time management skills, self-reliance, regular engagement and communication with the lecturer. Additionally, students could suffer from the absence of vital personal interactions, not only with lecturers, but also with colleagues that participate in the module [7,8]. On the other hand, students are offered a more flexible learning process, that is especially convenient when they study several subjects simultaneously and if they are required to combine studying with professional work. The e-learning system can improve communication between lecturers and students [9]. As many corporations move towards online activities, future graduates should be able to develop skills that will help them in their future jobs. Skills such as conscientiousness, independence, and creativity are the key ones. Moreover, graduates will be required to continue education and self-education in order to maintain their competences at a high level [10].

For higher education institutions, embedding e-learning into the curriculum means investing in IT infrastructure and up-to-date teaching tools. It is also necessary to develop training programs for the staff. However, at the same time, the institution can reduce the cost of premises and increase the number of students enrolled in courses and programs, as there is no limit on the number of people who can join online classes [11].

Lecturers who use the e-learning method will have to increase their online availability, but they will also be required to re-design, develop and implement appropriate online materials. Another problem is that, although some lecturers have an excellent knowledge of academic subjects, they may not have the relevant skills to deliver e-learning modules. Many researchers emphasize that there are great benefits if e-learning is properly applied [9,12–16], as this type of delivery allows the lecturer to have more flexible teaching hours and to work from home on many occasions [17].

Some disciplines may require face to face clarifications and explanations [11]. Very often, the lecturers involved in those courses believe that face to face interaction with the student makes the learning process much easier and more efficient. In these cases, the e-learning method might be less effective than the traditional method of learning. Some research studies have argued that e-learning is more appropriate in social science and humanities, than in fields such as medical science, mathematics, chemistry, and pharmacy, where there is the need to develop practical skills [18]. Purely scientific fields, which use more practical approaches, may need to modify the e-learning method to suit their courses.

The effectiveness of learning is understood as achieving learning outcomes set out in the course outline, according to the National Qualifications Framework or the course syllabus, which, in reality, means student test score. For academic teachers, learning effectiveness means perceived learner satisfaction. This approach is consistent with Zhang et al. [16]

## 3. Educators' Approach to E-Learning

There is a lot of research that looks at e-learning in higher education. However, most of it concerns the use of e-learning from the students' perspective [19–22]. Klimczak highlights the various benefits of blended learning [23]. E-learning tools can prove to be especially beneficial when dealing with large cohorts of students, as they simplify the assessment process and make it more efficient and less time-consuming. Students that participate in online course delivery are forced to engage in online activities on a regular basis, which is sometimes difficult with courses offered on campus. However, it is important to consider how e-learning modules are perceived by lecturers. It is necessary to establish which factors play a significant role for the educators, which elements help them to run efficient online modules and which obstacles need to be overcome.

Numerous factors determine the adaptability of the e-learning approach by lecturers [24]. Some lecturers are reluctant to introduce this form of education, due to lack of experience in using information technology and lack of adequate support from the IT department. This new teaching approach also requires different forms of communication with students, which for many lecturers may cause discomfort and enforce changes in their existing habits. Often lecturers' attitudes, as well as psychological considerations, influence the form of teaching and how courses are delivered. Researchers very often stress the impact of lecturers' experience with information technology, modern methods of communication and awareness of the impact of technology on the efficiency of the teaching process. Those who apply e-learning in one of their modules are more likely to apply this approach in the subsequent courses they teach.

A substantial number of academics do not apply e-learning in their teaching, because they do not fully understand what it involves and very often overestimate the difficulties and risks associated with this form of teaching.

The strong factor that affects the use of the e-learning teaching approach is the simplicity of delivering e-modules, but the most important determinants of the impacts on the adoption process are the awareness and knowledge of information technology and previous experience of delivering online courses [24–26]. These two major factors make the development and implementation of e-learning much easier. Nevertheless, lecturers who have never been involved in e-learning very often perceive its development and delivery as more difficult and complicated than it is in practice.

Jebeile surveyed Australian lecturers that adopted web-based technology in their teaching, where they looked at the evaluation of online learning through the use of Internet communication [27]. As important factors determining the use of e-learning, among other things, lecturers pointed out the higher quality, efficiency and effectiveness of online activities. When using the online learning platform, students can make a clear, objective and measurable evaluation of learning activities. It is clear, that the courses that can train academic staff in designing and delivering online classes can reduce the barriers and negative attitudes of lecturers in the field of teaching online. There is the possibility of learning a valuable lesson from experienced users as well. Due to the key importance of e-learning training, they should be conducted in individual schools, or they should be organized by the Ministry of Education for all teachers [28].

Finally, a very important factor that impacts the adoption of online learning is the ease of its use. However, for many lecturers, who are not familiar with this new teaching approach, several concerns can increase resistance to the process. For example, the complexity of the software can aggravate anxiety [26,29–31]. This can be minimized by creating e-learning centers, that could help better prepare students and lecturers to adapt to the e-learning process. The implementation of user-friendly software and platforms also has a significant impact on the success of the whole process [7]. The ease of using IT solutions in e-learning classes and the perceived usefulness of e-learning classes are key factors determining the acceptance of this form of teaching by lecturers [32–34].

Embedding e-learning does not always result in students being more engaged, knowledgeable and developing better skills, and tutors' beliefs strongly impact the undertaking and design of blended learning modules [35].

The most important factors determining the usefulness of e-learning are the learning environment, the way in which course content is delivered and lecturers' attitude to e-learning [36]. These factors are important for lecturers, students, and faculties, and can be used as evaluation criteria for this form of module delivery. The learning environment includes learning spaces (lecture theatres, classrooms), teaching materials, appropriate computer software, etc. A friendly learning environment increases course effectiveness for both students and lecturers. The way in which course content is delivered also impacts on module efficiency. It should be considered whether e-learning classes should be implemented interchangeably with traditional delivery. What is more, it is worth discussing whether the degree of difficulty of the specific content should be evenly distributed between traditional classes and e-learning ones.

Great significance has been linked to lecturers' role, which has changed, as they are no longer only experts who deliver a certain type of knowledge, but also individuals who help solve problems related to e-learning facilities, as well as course guides [37]. The lecturer in this process is considered a problem-solver, or the person who causes problems. One of the factors that can help facilitate the e-learning process is the possibility of obtaining professional and technical assistance and support during the implementation of the e-learning process, for both lecturers and students. Diverse technical problems can cause lecturers additional work and instead of focusing on the teaching process, they become more administrators and facilitators of the learning process.

The financial constraint has been mentioned as one of the most essential barriers for academics when using online technology. The unavailability of appropriate hardware and software necessary for the efficient delivery of online modules is due to the fact that some software can be rather expensive and, therefore, inaccessible to some institutions [38].

It must be stressed that the new generation of students demands the use of innovative forms of teaching, hence some lecturers try to meet these demands and embed various forms of e-learning in their course delivery. Lecturers who offer e-learning classes are aware of the added value of this form of teaching. By engaging in online courses, students can gain the opportunity to develop many additional skills needed for their future professional career [39]. These include additional cognitive skills, such as the ability to organize a variety of information, to obtain information on their own, to develop methods of creative thinking, formulate different points of view, discuss, analyze and solve a variety of problems. The opportunity to develop further skills can also be a reason for selecting a particular form of teaching by the lecturer. Mahdizadeh et al. show that 2/3 of academics based their decision to use e-learning in their teaching process on their belief that they provided their students with added value [40]. They also emphasized some essential elements determining the e-learning adoption process, which included practical use of the system, the easiness or difficulty of its use, as well as the amount of time the whole process takes. It is not without significance that if lecturers had previous experience of creating e-learning classes, they were keener to use this type of approach for other forms of teaching.

Willingness to use e-learning also depends on the already mentioned lecturers' "attitudes" and their personal experience in this matter. Hence, they should be encouraged by the university to use this new form of delivery in their teaching. Academics may not necessarily be keen to change their habits and often have a negative attitude to a particular method of delivering activities (e-learning). Raman et al. also draw attention to the importance of the right attitudes of lecturers when they decide on the use of e-learning classes. Lecturers can use their habits while implementing e-learning.; whether they are more or less open to changes, they can introduce them with greater or lesser efficiency, and their preferred teaching style is also important [41]. Acceptance by e-learning by lecturers may also have cultural reasons [4,42].

Lecturers' characteristics and their involvement in the online teaching process have very strong impacts on the effectiveness of e-learning [43]. At the same time, it is important for lecturers to take a more positive attitude. They should be more open to using technology in their teaching, including online classes. The factors that determine the level of acceptance of this form of teaching among lecturers ought to be considered. Therefore, more effort should be put into improving the awareness of the learning environment, in particular, various platforms and software that can facilitate the learning process. Teo indicated that lecturers should complete the necessary training prior to getting involved in e-learning [26]. They should be familiar with the relevant tools, which will then be used regularly during the online classes with the students.

One of the reasons why lecturers may be discouraged from delivering classes in the e-learning form can also be a lack of acceptance of commonly used teaching methods for online classes [44]. The course design needs to be closely linked with interactive tools that can be embedded within the course and the response to students' needs. Lecturers must be open to the evaluation of their courses, especially to avoid the emotional and cognitive disconnection between them and the students.

Lecturers should, therefore, verify whether the information provided to students is clear and precise, and if feedback from the tutors is transmitted regularly and without delay.

One of the factors affecting the perception of online activities is the potential time saving for lecturers [45]. The online delivery allows classes to be taught more efficiently and enables better time management and often time savings, which is a great benefit for lecturers. Some of the lecturers indicate that one of the real benefits of e-delivery is the increasing ability to manage their own time efficiently. Teaching online is the most desirable in situations where teaching space such as in lecture theatres, laboratories, IT rooms, and tutorial rooms is limited.

On the other hand, delivering classes online involves a greater investment of time than conducting classes on campus and often results in the need for the lecturer to be available for several hours a day, seven days a week [46]. Moreover, the flexible, open virtual character of online classes can increase the number of questions that need to be answered during e-sessions. Hence, the frequency of interaction between the lecturer and course participants grows and the duration of the e-class is prolonged. It should also be stressed that lecturers need extra time to get to understand information technology and its proficient use for creating and administering e-learning sessions. They need to allocate time for online communication and other activities related to the e-course [47]. As lots of lecturers very often don't use other people's experience in delivering e-courses, they require additional time to design and launch the e-learning course.

There is a lack of awareness of the importance of e-learning form of teaching among the lecturers themselves and many academics highlight the fundamental problems when designing and implementing the e-learning approach [48]. This calls for greater support provided by universities in the field of knowledge transfer and new ways of effective course delivery via the Internet. Sadik's results confirm that a significant part of faculty members have limited competence in the pedagogy for online design and delivery [48]. Lecturers believe that this form of teaching is useful and can be beneficial and efficient if designed and executed properly. However, they raise concerns about additional knowledge, skills and training that are required, and extra time that needs to be invested. Furthermore, not all institutions can afford the sophisticated software that is needed to deliver good quality and interactive sessions.

Therefore, it seems necessary to provide proper support for lecturers who want to be involved in e-learning courses and to supply adequate and reliable infrastructure, in order to increase the number of courses offered in an e-learning module. There is a need for systemic support for lecturers engaged in the implementation of online courses, as without relevant administrative and technical support a large amount of academics are skeptical about using e-learning for their courses [49]. Universities can increase the number of e-learning courses if the academic staff do not have to face a variety of problems related to the implementation and administration of online modules [24]. An adequate IT infrastructure needs to be provided, to reduce concerns about the essential change in the form of teaching, and a smooth transition process should be offered.

## 4. Empirical Findings

### 4.1. The Purpose and Method of Research

As adopting e-learning in higher education is an important process, in the paper the attitude of Polish educators towards the implementation of e-learning in the teaching of accounting modules will be examined. The main purpose of the survey was to investigate their opinions on the benefits and shortcomings associated with the e-courses and to find out about their views on the e-courses in comparison with traditional classes. The primary source of data was a questionnaire conducted among the accounting lecturers employed by the leading Polish Economic Universities, such as the Cracow University of Economics, Wroclaw University of Economics, Poznan University of Economics and Business, University of Lodz, Kozminski University, the University of Economics in Katowice,

University of Szczecin, University of Gdansk, Nicholas Copernicus University in Torun, and Warsaw School of Economics.

The survey was carried out in two stages. The first stage of the research focused on distributing printed questionnaires among the participants of the Polish Accounting Departments Annual Conference, which took place on September 28–30, 2016 in Katowice. Such reunions are held every year and participants include teaching and academic staff of the most important scientific centers in Poland. The paper survey was used, as it guarantees a high response to the questionnaire. In the second stage, the printed questionnaire was converted to an online survey. The link to the online survey was circulated via e-mail to the staff of various higher education institutions, whose email addresses were included in the database held by the Department of Financial Accounting of the Cracow University of Economics. Only the people who did not participate in the printed questionnaire were invited to complete the online survey.

The questionnaire consisted of 24 questions, divided into two groups. The first group concerned the benefits and problems related to the use of e-learning (12 questions) and the second group related to the differences between e-learning and traditional delivery (12 questions). With regards to the first group of questions, respondents could indicate the three most important to them out of five proposed benefits and also three out of five difficulties. Their importance was not differentiated. The answers to the second group of questions were provided on a scale from 1–9, where 1 was described as "the easiest", "the least" or "the lowest" and 9 as "the most difficult", "the most" or "the highest". The answers "the easiest/ the most difficult" related to the first two questions, "the least/the most" to the next six, and "the lowest /the highest" to the last four. A similar Likert scale questionnaire is also used in the literature [17,50].

### 4.2. Research Sample

In total, 79 respondents participated in the survey, including 41 females and 38 males. Nineteen respondents implemented e-learning. They conducted courses, in which the relation between traditional and e-learning hours was on average 75%/25%. They delivered e-learning courses for six years on average, however, there was one person with 15-year experience in this matter. The courses were mainly offered to full-time students. In Table 1, the respondents' work experience and teaching load are presented.

**Table 1.** Respondents' work experience and teaching load.

| Respondents—Females | | | | |
|---|---|---|---|---|
| | **Maximum** | **Minimum** | **Mean** | **Median** |
| Work experience (years) | 40 | 1 | 16.3 | 16.5 |
| Teaching load per academic year (teaching hours) * | 600 | 46 | 326 | 300 |
| Respondents—Males | | | | |
| | Maximum | Minimum | Mean | Median |
| Work experience (years) | 40 | 1 | 16.0 | 15.5 |
| Teaching load (teaching hours) * | 800 | 60 | 396 | 350 |

* one teaching hour is 45 minutes long; Source: Authors' own elaboration.

There was no difference in job seniority between males and females participating in the study, however, the teaching experience of both groups varied from 1 to 40 years (Table 1). The maximum teaching workload indicated by males is 800 teaching hours per academic year, so it is higher than in the case of females (600 hours). The mean teaching load, which is higher for males than females, is equal to or above 300 hours in both cases and might be perceived as relatively high in the Polish academic environment.

*4.3. Survey Results*

4.3.1. Benefits and Shortcomings Associated with E-Courses

　　In the survey, academics' perception of benefits and drawbacks associated with e-courses were investigated. It should be noted that educators who have already been using e-learning in their teaching answer these questions based on their own experiences. The other respondents based their responses on their projections regarding these issues.

　　The respondents indicated (see Table 2) that the most important benefit of delivering e-courses is an enhancement of the efficiency of the teaching process (sixty respondents, i.e. 76%). Secondly, the flexibility of the teaching process was indicated (fifty-six educators, i.e. 71%). This was followed by time-saving (forty respondents, i.e. 51%) and keeping pace with changing technology (thirty-eight participants, i.e. 48%). More effective communication with students was chosen as the least beneficial aspect of e-learning (twenty-eight teachers, i.e. 35%), although lots of respondents still admitted that e-learning improves this part of the teaching process.

**Table 2.** Benefits associated with the e-courses.

| Benefits | Responses (%) |
|---|---|
| Better efficiency of the teaching process (implementation of learning outcomes set in course outline/course syllabus) | 76 |
| The possibility of conducting teaching at any time and from anywhere (convenient for me) | 71 |
| Time-saving (no need to come to campus to deliver teaching, easier sharing of teaching materials (online)) | 51 |
| Satisfaction with use of innovative teaching approach (adapted to the changes of technical/technological progress) | 48 |
| The more efficient communication process with students (e-mail, announcements, e-consultations) | 35 |

Source: Authors' own elaboration.

　　A large amount of work associated with designing and updating course materials was seen by respondents as the main difficulty associated with e-courses (Table 3). Fifty-three of the participants (67%) indicated that this is a constraint that can put educators off embedding e-learning in their courses. Technical problems that might occur during course design and course delivery phases were seen as a difficulty by forty of respondents (51%). Mechanization of the learning process and the need to solve problems reported by students were not so important. They were indicated by thirty-six (46%) and thirty (38%) educators, respectively. Twenty-nine respondents (37%) specified the need to spend additional time on conducting online activities with students as an important difficulty.

**Table 3.** Difficulties associated with the e-courses.

| Difficulties | Responses (%) |
|---|---|
| A large amount of work associated with designing and updating course materials | 67 |
| The necessity to overcome technical problems during course preparation and course delivery | 51 |
| A sense of excessive mechanization of the learning process (limited opportunity to establish closer, personal relationship with students) | 46 |
| The necessity of solving technical problems reported by students | 38 |
| The necessity to dedicate time to conduct online activities with students (e.g. participation in discussions, answering questions via e-mail, managing e-forums, e-consultations) | 37 |

Source: Authors' own elaboration.

The respondents have also been asked to compare e-learning courses with traditional ones (Table 4). In order to make the presentation of the results more transparent, the responses were grouped into three main categories: (1) less/lower/smaller (answers: 1–4); (2) same (answer: 5); (3) more/higher/greater (answers 6–9). Preparation and delivery of e-learning courses are perceived by respondents as more difficult (sixty-eight respondents, i.e. 86%, and thirty-three respondents, i.e. 42% respectively) and at the same time, more time consuming (sixty-two teachers, i.e. 79%) than in the case of traditional classes. It is worth noting that only three participants, i.e. 4%, thought of e-learning as less time-consuming.

**Table 4.** Evaluation of e-courses in comparison to traditional ones.

| Questions | Responses (in %) | | |
|---|---|---|---|
| Preparation of e-learning course in comparison with traditionally delivered classes is: | less difficult | same | more difficult |
| | 1 | 13 | 86 |
| E-learning delivery in comparison with traditional delivery is: | less difficult | same | more difficult |
| | 35 | 23 | 42 |
| In order to prepare e-learning classes as opposed to traditional classes, the lecturer needs: | less time | same time | more time |
| | 4 | 17 | 79% |
| Delivering e-learning classes in comparison with traditional classes takes: | less time | same time | more time |
| | 37 | 35 | 28 |
| Studying for e-learning classes in comparison with traditional classes takes students: | less time | same time | more time |
| | 29 | 35 | 36 |
| Degree of regularity with which students learning during e-learning classes in comparison to traditional classes is: | smaller | same | greater |
| | 45 | 28 | 27 |
| The range of material possible to be transferred to students during e-learning course in comparison with traditional delivery is: | smaller | same | greater |
| | 18 | 32 | 50 |
| Student activity during e-learning session in comparison with traditional delivery is: | lower | same | higher |
| | 43 | 23 | 33 |
| Level of development of students' social competences during e-learning classes in comparison with traditional classes is: | lower | same | higher |
| | 66 | 21 | 14 |
| The danger of students dishonest behavior during e-learning classes in comparison with traditional classes is: | smaller | same | greater |
| | 4 | 8 | 88 |
| The possibility of verifying students' progress during e-learning classes in comparison with traditional classes is: | smaller | same | greater |
| | 51 | 20 | 30 |
| Effectiveness of e-learning teaching comparison with traditional teaching is: | lower | same | higher |
| | 48 | 23 | 30 |

Source: Authors' own elaboration.

According to the respondents (Table 4), academics have to spend less time delivering e-learning classes (twenty-nine respondents, i.e. 37%), but students need to spend more time studying (twenty-eight participants, i.e. 35%). However, it should be noted that in both cases there were almost no differences between the percentage of neutral responses ("same") and responses "more" or "less". As indicated by the survey participants, students' work during e-classes requires more systematic approach (45%) and students are also able to get familiar with a wider range of topics (forty educators, i.e. above 50%). Social competencies, such as the ability to organize working time, teamwork, communication, are an important part of the learning process. Lecturers indicated that the degree of development of these competencies during e-learning classes is lower in comparison with traditional classes (fifty-two respondents, i.e. 66% chose this option). The same pattern can also be observed in relation to students' activity during the course, since thirty-four teachers, i.e. 43%, agreed that it is lower in comparison to the traditional delivery. What is more, the possibility of verifying the progress of students' learning during e-learning classes in comparison with traditional courses is smaller (forty educators, i.e. 51 %).

Most of the respondents (seventy participants, i.e. 88%) indicated the higher probability of students' dishonest behavior during e-learning classes as a differentiating factor between traditional and e-courses. Examples of such behavior include; joint work on tasks that have been assigned as individual tests, the use of other students' work and unauthorized aids, the lack of contribution and participation in group activities, etc.

Respondents indicated that the effectiveness of e-learning teaching in comparison with traditional teaching is lower (thirty-eight respondents, i.e. 48% of them did so). The results are inconsistent with the previous statement, where sixty teachers, i.e. 76%, indicated the efficiency of the teaching process as the most important benefit for the educator.

### 4.3.2. Econometric Modeling

In order to perform a more thorough evaluation of the relationship between the overall evaluation of e-learning and other determinants, a MANOVA and regression analysis has been carried out. Empirical analysis has been performed, based on the survey described in the previous part of the paper. The content of the survey consists of categorical questions divided into two groups. The one group concerned the benefits and problems related to the use of e-learning (10 questions) and responses to them may be presented as dummy variables (zero-one). The second group related to the differences between e-learning and traditional delivery (12 questions). The responses to them were indicated on a scale from 1 to 9.

This study firstly investigates the perception of the effectiveness of e-learning techniques within two groups of academics—the ones who use e-learning and the others. Secondly, we analyze factors influencing the effectiveness of e-learning versus traditional methods, as perceived by lecturers/academics. Thirdly, in the research, the use of e-learning techniques for teaching accounting subjects was examined, and in particular, the reasons behind this decision, namely, the perceived benefits and shortcomings of blended learning in relation to traditional teaching. Table 5 presents the content of the questions, which were used as variables in the empirical models.

**Table 5.** The first group of categorical variables (in scale 1 to 9).

| Variable Code | Description |
|---|---|
| Y | Effectiveness of e-learning teaching in comparison with traditional teaching |
| P01 | The difficulty of preparation of e-learning course in comparison with traditionally delivered classes |
| P02 | The difficulty of e-learning delivery in comparison with traditional delivery |
| P03 | The time required by the lecturer to prepare e-learning classes in comparison with time spent when preparing traditional classes |
| P04 | The time spent by the lecturer to deliver e-learning classes in comparison with time spent when delivering traditional classes |
| P05 | The time dedicated by students to study for e-learning classes in comparison with traditional classes |
| P06 | The regularity of students' learning during e-learning classes in comparison to traditional classes |
| P07 | The range of material possible to be transferred to students during e-learning course in comparison with traditional delivery |
| P08 | Student activity during e-learning session in comparison with traditional delivery |
| P09 | Development of students' social competences during e-learning classes in comparison with traditional classes |
| P10 | The danger of students' dishonest behavior during e-learning classes in comparison with traditional classes |
| P11 | The possibility of verifying the progress of students' learning during e-learning classes in comparison with traditional classes |

Source: Authors' own elaboration.

Based on the literature review and researchers' own experience as lecturers of accounting subjects, the following three hypotheses have been established:

**Hypothesis 1.** *The perception of the effectiveness of e-learning techniques differs between the group of academics, who already use it and the group, which have not decided to use it.*

**Hypothesis 2.** *The perceived effectiveness of e-learning teaching in comparison with traditional teaching is dependent on the perceived advantages and disadvantages of e-learning.*

**Hypothesis 3**. *The decision to use e-learning methods in teaching accounting subjects is based on the perceived advantages and disadvantages of e-learning.*

Hypotheses are verified by a statistical analysis of data acquired in an opinion poll. Table 6 displays the summary statistics of the first group of variables derived from the questionnaire.

**Table 6.** Summary statistics of first group variables.

| Variables | Obs. | Mean | St. Dev. | Min. | Max. |
|---|---|---|---|---|---|
| **Y** | 71 | 4.507 | 1.904 | 1 | 9 |
| **P01** | 72 | 6.903 | 1.291 | 4 | 9 |
| **P02** | 71 | 5.183 | 1.807 | 1 | 9 |
| **P03** | 72 | 6.819 | 1.621 | 2 | 9 |
| **P04** | 71 | 5.070 | 1.467 | 2 | 9 |
| **P05** | 72 | 5.389 | 1.765 | 1 | 9 |
| **P06** | 71 | 4.577 | 2.102 | 1 | 9 |
| **P07** | 72 | 5.736 | 1.839 | 1 | 9 |
| **P08** | 69 | 4.681 | 1.859 | 1 | 9 |
| **P09** | 73 | 3.699 | 1.823 | 1 | 8 |
| **P10** | 72 | 7.139 | 1.513 | 1 | 9 |
| **P11** | 71 | 4.465 | 2.110 | 1 | 9 |

Source: Authors' own elaboration.

To select variables for the empirical model, the strength of the correlation between variables has been tested. In the correlation analysis, as presented in Table 7, a strong interdependence between variable P01 and P03 can be observed and between P11 and respectively Y, P06, and P08.

**Table 7.** Correlation matrix between variables P01–P11 and Y.

| Variables | Y | P01 | P02 | P03 | P04 | P05 | P06 | P07 | P08 | P09 | P10 | P11 |
|---|---|---|---|---|---|---|---|---|---|---|---|---|
| **Y** | 1.00 | | | | | | | | | | | |
| **P01** | 0.08 | 1.00 | | | | | | | | | | |
| **P02** | −0.08 | 0.33 | 1.00 | | | | | | | | | |
| **P03** | 0.07 | 0.78 | 0.21 | 1.00 | | | | | | | | |
| **P04** | 0.08 | 0.38 | 0.38 | 0.40 | 1.00 | | | | | | | |
| **P05** | 0.12 | −0.10 | −0.32 | −0.15 | 0.12 | 1.00 | | | | | | |
| **P06** | 0.60 | 0.04 | 0.03 | 0.01 | 0.10 | 0.36 | 1.00 | | | | | |
| **P07** | 0.46 | −0.01 | 0.13 | 0.01 | 0.16 | 0.21 | 0.44 | 1.00 | | | | |
| **P08** | 0.60 | 0.25 | 0.18 | 0.21 | 0.09 | 0.18 | 0.59 | 0.54 | 1.00 | | | |
| **P09** | 0.63 | 0.11 | 0.05 | 0.08 | 0.14 | 0.04 | 0.37 | 0.45 | 0.56 | 1.00 | | |
| **P10** | −0.27 | 0.13 | −0.16 | 0.15 | 0.06 | 0.05 | −0.17 | −0.12 | −0.19 | −0.20 | 1.00 | |
| **P11** | 0.66 | −0.06 | 0.02 | −0.06 | 0.06 | 0.25 | 0.62 | 0.40 | 0.58 | 0.44 | −0.24 | 1.00 |

Source: Authors' own elaboration.

In order to test whether the perception of the effectiveness of e-learning techniques differs between the two groups of academics, we employ MANOVA analysis (see Table 8). The results show that there is a significant difference between lecturers who use and don't use e-learning techniques, in terms of the perception of advantages and disadvantages. As a grouping variable, we are using E-LEARN, and P01, P02, . . . , P10 as dependent variables. We excluded P11, due to a high correlation with Y, P06 and P08 variables.

In order to test the second hypothesis, we employ linear regression. The results (see Table 9) show that variables P02, P06, P09, P11 are statistically significant determinants of the dependent variable (at a significance level 0.05). In this case, we excluded the P03 variable, due to a high correlation with the P01 variable.

**Table 8.** Results of the MANOVA.

| Source | | Number of Obs = 68<br>W = Wilks' lambda<br>P = Pillai's trace<br>Statistic | df | F(df1, | L = Lawley-Hotelling trace<br>R = Roy's largest root<br>df2) = | F | Prob > F | |
|---|---|---|---|---|---|---|---|---|
| E_LEARN | W | 0.7377 | 1 | 10.0 | 57.0 | 2.03 | 0.0470 | e |
| | P | 0.2623 | | 10.0 | 57.0 | 2.03 | 0.0470 | e |
| | L | 0.3556 | | 10.0 | 57.0 | 2.03 | 0.0470 | e |
| | R | 0.3556 | | 10.0 | 57.0 | 2.03 | 0.0470 | e |
| Residual | | 66 | | | | | | |
| Total | | 67 | | | | | | |
| e = exact, a = approximate, u = upper bound on F | | | | | | | | |

**Table 9.** Results of the linear regression.

| Independent Variable | | Coeff | Std. Error | t-Statistic | *p*-Value | (95% Conf. Interval) | |
|---|---|---|---|---|---|---|---|
| (Constant) | | 0.0694 | 1.2995 | 1.59 | 0.12 | −0.5339 | 4.6726 |
| P01 | | 0.1601 | 0.1374 | 1.17 | 0.25 | −0.1151 | 0.4354 |
| P02 | | −0.2662 | 0.1006 | −2.65 *** | 0.01 | −0.4678 | −0.0646 |
| P04 | | 0.0643 | 0.1204 | 0.53 | 0.60 | −0.1768 | 0.3054 |
| P05 | | −0.1782 | 0.1030 | −1.73 * | 0.09 | −0.3844 | 0.0281 |
| P06 | | 0.2067 | 0.0978 | 2.11 ** | 0.04 | 0.0107 | 0.4027 |
| P07 | | 0.1068 | 0.0996 | 107 | 0.29 | −0.0929 | 0.3064 |
| P08 | | 0.0683 | 0.1262 | 0.54 | 0.59 | −0.1846 | 0.3211 |
| P09 | | 0.3026 | 0.1054 | 2.87 *** | 0.01 | 0.0915 | 0.5136 |
| P10 | | −0.1392 | 0.1023 | −1.36 | 0.18 | −0.3441 | 0.0658 |
| P11 | | 0.2988 | 0.0978 | 3.05 *** | 0.00 | 0.1028 | 0.4947 |
| Prob > F | = | 0.0000 | | | | | |
| R-squared | = | 0.6713 | | | | | |
| Adjusted R-squared | = | 0.6126 | | | | | |
| Root MSE | = | 1.1836 | | | | | |

Notes: *** Significance at 1% level, ** significance at 5% level, * significance at 10% level; Source: Authors' own elaboration.

The outcome of the analysis suggests that the positive impact on the perceived effectiveness of e-learning teaching is statistically determined by perceived:

- easiness of e-learning delivery in comparison with traditional methods (P02, negative relation);
- more regular learning process during e-learning class in comparison with traditional classes (P06, positive relation);
- a greater degree of development of students' social competences during e-learning classes in comparison with traditional classes (P09, positive relation);
- a more effective process of verification of students' progress in the case of e-learning teaching (P11, positive relation).

In the case of other variables, the study did not find any evidence of statistical significance, therefore, our study provides evidence supporting the second hypothesis.

The third hypothesis is related to the decision to use or not to use e-learning techniques in teaching accounting subjects at the university level. The potential determinants of this decision are the perceived advantages and disadvantages of e-learning teaching. The dependent variable, in this case, is a dummy variable, so it imposes a necessity to apply a logit regression. This hypothesis was verified by using two groups of variables. The first one was presented in Table 5 and the second group is presented below (see Table 10).

**Table 10.** Dummy variables of the second group of variables

| Variable Code | Description |
|---|---|
| E_LEARN | The decision to teach or not to teach accounting subjects in e-learning mode |
| A1 | the possibility of conducting teaching at any time and from anywhere (convenient for me) |
| A2 | time-saving (no need to come to Campus to deliver teaching, easier sharing of teaching materials (on-line)) |
| A3 | satisfaction with the use of innovative teaching approach (adapted to the changes of technical/technological progress) |
| A4 | the more efficient communication process with students (e-mail, announcement, e-meetings) |
| A5 | better efficiency of the teaching process (implementation of learning outcomes set in course outline/course syllabus) |
| D1 | a large amount of work associated with designing and updating course materials |
| D2 | the necessity to overcome technical problems during course preparation and course delivery |
| D3 | the necessity of solving technical problems reported by students |
| D4 | a sense of excessive mechanization of the learning process (limited opportunity to establish closer, personal relationship with students) |
| D5 | the necessity to dedicate time to conducting online activities with students (e.g. participation in discussions, answering questions via e-mail, managing e-forums, e-meetings) |

Source: Authors' own elaboration.

Table 11 presents the summary statistics of the second group of variables.

**Table 11.** Summary statistics of the second group of variables

| Variables | Obs. | Mean | St. Dev. | Min. | Max. |
|---|---|---|---|---|---|
| E_LEARN | 79 | 0.2405 | 0.4301 | 0 | 1 |
| A1 | 79 | 0.7089 | 0.4572 | 0 | 1 |
| A2 | 79 | 0.5063 | 0.5032 | 0 | 1 |
| A3 | 79 | 0.4810 | 0.5028 | 0 | 1 |
| A4 | 79 | 0.3544 | 0.4814 | 0 | 1 |
| A5 | 79 | 0.7594 | 0.4301 | 0 | 1 |
| D1 | 79 | 0.6709 | 0.4729 | 0 | 1 |
| D2 | 79 | 0.5063 | 0.5032 | 0 | 1 |
| D3 | 79 | 0.3797 | 0.4884 | 0 | 1 |
| D4 | 79 | 0.4557 | 0.5012 | 0 | 1 |
| D5 | 79 | 0.3671 | 0.4851 | 0 | 1 |

Source: Authors' own elaboration.

The empirical analysis started with the first group of determinants. Firstly, the correlation between the E_LEARN variable and P01-P11 has been tested (see Table 12).

**Table 12.** Correlation matrix between E_LEARN and P01–P11 variables.

| Variable | E_LEARN | P01 | P02 | P03 | P04 | P05 | P06 | P07 | P08 | P09 | P10 | P11 |
|---|---|---|---|---|---|---|---|---|---|---|---|---|
| **E_LEARN** | 1.00 | | | | | | | | | | | |
| **P01** | 0.01 | 1.00 | | | | | | | | | | |
| **P02** | 0.32 | 0.33 | 1.00 | | | | | | | | | |
| **P03** | 0.04 | 0.78 | 0.21 | 1.00 | | | | | | | | |
| **P04** | 0.18 | 0.38 | 0.38 | 0.40 | 1.00 | | | | | | | |
| **P05** | 0.11 | −0.10 | −0.32 | −0.15 | 0.12 | 1.00 | | | | | | |
| **P06** | 0.15 | 0.04 | 0.03 | 0.01 | 0.10 | 0.36 | 1.00 | | | | | |
| **P07** | 0.00 | −0.01 | 0.13 | 0.01 | 0.16 | 0.21 | 0.44 | 1.00 | | | | |
| **P08** | 0.02 | 0.25 | 0.18 | 0.21 | 0.09 | 0.18 | 0.59 | 0.54 | 1.00 | | | |
| **P09** | 0.02 | 0.11 | 0.05 | 0.08 | 0.14 | 0.04 | 0.37 | 0.45 | 0.56 | 1.00 | | |
| **P10** | −0.22 | 0.13 | −0.16 | 0.15 | 0.06 | 0.05 | −0.17 | −0.12 | −0.19 | −0.20 | 1.00 | |
| **P11** | 0.22 | −0.06 | 0.02 | −0.06 | 0.06 | 0.25 | 0.62 | 0.40 | 0.58 | 0.44 | −0.24 | 1.00 |

Source: Authors' own elaboration.

Again, the results show a strong correlation between P01 and P03, and again, the P03 variable, due to a very weak correlation to the E-LEARN variable, has been eliminated. The next step that was implemented was a logit regression (see Table 13).

**Table 13.** Logit regression of E_LEARN and the first group of variables

| Independent Variable | | Coeff | Odds Ratio | Std. Error | z-Statistic | P > | z | | [95% Conf. Interval] | |
|---|---|---|---|---|---|---|---|---|
| (Constant) | | −2.6347 | 0.7174 | 2.8679 | −0.92 | 0.36 | −8.2556 | 2.9862 |
| P01 | | −0.3526 | 0.7028 | 0.3062 | −1.15 | 0.25 | −0.9528 | 0.2475 |
| P02 | | 0.7183 | 2.0510 | 0.2751 | 2.61 *** | 0.01 | 0.1792 | 1.2575 |
| P04 | | 0.1896 | 1.2088 | 0.2673 | 0.71 | 0.48 | −0.3343 | 0.7136 |
| P05 | | 0.2737 | 1.3148 | 0.2483 | 1.1 | 0.27 | −0.2131 | 0.7604 |
| P06 | | −0.0026 | 0.9974 | 0.2391 | −0.01 | 0.99 | −0.4712 | 0.4659 |
| P07 | | −0.5789 | 0.5605 | 0.3082 | −1.88 * | 0.06 | −1.1829 | 0.0251 |
| P08 | | 0.3285 | 1.3889 | 0.2971 | 1.11 | 0.27 | −0.2538 | 0.9108 |
| P09 | | −0.0814 | 0.9218 | 0.2296 | −0.35 | 0.72 | −0.5313 | 0.3685 |
| P10 | | 0.2543 | 0.7755 | 0.2147 | −1.18 | 0.24 | −0.6750 | 0.1664 |
| P11 | | 0.3073 | 1.3598 | 0.2500 | 1.23 | 0.22 | −0.1827 | 0.7974 |
| LR chi2(10) | = | 21.17 | | | | | | |
| Prob > chi2 | = | 0.0199 | | | | | | |
| PseudoR2 | = | 0.52.99 | | | | | | |
| Log Likelihood | = | −29.3672 | | | | | | |

Notes: *** Significance at 1% level, ** significance at 5% level, * significance at 10% level; Source: Authors' own elaboration.

Based on the likelihood ratio chi-square, it should be stressed that our model as a whole fits significantly better than an empty model. Variable P02 and P07 are the only ones, which are statistically significant. The interpretation is based on the odds ratio and the results can be read as follows:

- the more difficult the perceived e-learning delivery, the more chance there is that the academic teacher will undertake e-learning teaching. So, the more demanding the didactic method, the more inclined the lecturers are to apply e-learning,
- the greater the range of material possible to be transferred to students during the e-learning course in comparison to traditional delivery, the less chance there is that the teacher will choose e-learning.

The results did not appeal to us, and we wanted to extend our study in this area. Therefore, we tested the E_LEARN variable against the second group of variables. In this case, we again performed correlation analysis. The results (see Table 14) show no strong correlation between any pair of variables.

**Table 14.** Correlation matrix between E-LEARN and second group of variables

| Variable | E_LEARN | A1 | A2 | A3 | A4 | A5 | D1 | D2 | D3 | D4 | D5 |
|---|---|---|---|---|---|---|---|---|---|---|---|
| **E_LEARN** | 1.00 | | | | | | | | | | |
| **A1** | 0.17 | 1.00 | | | | | | | | | |
| **A2** | −0.10 | 0.20 | 1.00 | | | | | | | | |
| **A3** | 0.05 | 0.28 | 0.19 | 1.00 | | | | | | | |
| **A4** | 0.20 | −0.05 | 0.04 | 0.03 | 1.00 | | | | | | |
| **A5** | 0.18 | 0.88 | 0.21 | 0.36 | 0.05 | 1.00 | | | | | |
| **D1** | 0.27 | 0.03 | 0.01 | 0.14 | 0.24 | 0.11 | 1.00 | | | | |
| **D2** | −0.06 | 0.15 | 0.29 | 0.04 | 0.04 | 0.16 | 0.17 | 1.00 | | | |
| **D3** | −0.01 | 0.04 | 0.20 | 0.13 | 0.18 | 0.14 | 0.16 | 0.46 | 1.00 | | |
| **D4** | −0.10 | 0.31 | −0.06 | 0.09 | 0.01 | 0.28 | −0.12 | −0.16 | −0.09 | 1.00 | |
| **D5** | 0.00 | 0.31 | 0.17 | 0.16 | 0.04 | 0.24 | 0.09 | −0.14 | −0.27 | −0.01 | 1.00 |

Source: Authors' own elaboration.

Finally, we performed a logit regression (see Table 15). Based on the outcome of the analysis, it should be noted that the model as a whole fits significantly better than an empty model. The results provide evidence on the motivation of academic teachers of accounting subjects, and advise whether or not to use e-learning techniques in the teaching of accounting subjects and more specifically identify the following determinants:

- the perception of the communication process with students (A4) as more efficient increases the chances of the decision to use e-learning;
- a large amount of work associated with designing and updating course materials (D1) increases the chances of using e-learning techniques;
- technical problems (D2) are the main obstacles which discourage academics from e-learning education;
- a sense of excessive mechanization of the e-learning process decreases the chances of the decision to use e-learning.

**Table 15.** Logit regression of E_LEARN and the second group of variables.

| Independent Variable | | Coeff | Odds Ratio | Std. Error | z-Statistic | P > \|z\| | (95% Conf. Interval) | |
|---|---|---|---|---|---|---|---|---|
| (Constant) | | −2.5868 | 0.0753 | 1.0293 | −2.51 | 0.01 | −4.6043 | −0.5694 |
| A1 | | 2.0784 | 7.9922 | 1.5283 | 1.36 | 0.17 | −0.9170 | 0.7386 |
| A2 | | −0.5605 | 0.5709 | 0.6628 | −0.85 | 0.40 | −1.8596 | 0.7836 |
| A3 | | −0.1238 | 0.8836 | 0.6488 | −0.19 | 0.84 | −1.3953 | 1.1488 |
| A4 | | 1.1396 | 3.1257 | 0.6695 | 1.70 * | 0.09 | −0.1726 | 2.4519 |
| A5 | | 0.2248 | 1.2521 | 1.6393 | 0.14 | 0.89 | −2.9881 | 3.4378 |
| D1 | | 1.7271 | 5.6247 | 0.9007 | 1.92 * | 0.06 | −0.0383 | 3.4926 |
| D2 | | −1.5028 | 0.2225 | 0.8039 | −1.87 * | 0.06 | −3.0784 | 0.0729 |
| D3 | | −0.2743 | 0.7601 | 0.7642 | −0.36 | 0.72 | −1.7721 | 1.2236 |
| D4 | | −1.3453 | 0.2605 | 0.7357 | −1.83 * | 0.07 | −2.7871 | 0.0966 |
| D5 | | −1.0902 | 0.3362 | 0.7669 | −1.42 | 0.16 | −2.5933 | 0.4129 |
| LR chi2(10) | = | 19.30 | | | | | | |
| Prob > chi2 | = | 0.0366 | | | | | | |
| Pseudo R2 | = | 0.4532 | | | | | | |
| Log Likelihood | = | −33.9312 | | | | | | |

Notes: *** Significance at 1% level, ** significance at 5% level, * significance at 10% level; Source: Authors' own elaboration.

In both regressions, we used post-estimation diagnostics and statistical tests to detect specification errors in the models. Performed tests didn't detect any specification errors. We also used Hosmer and Lemeshow's goodness-of-fit test, which indicates that both models fit the data well.

Most notably, better communication with students is the main determinant motivating academics to invest in e-learning education. Better communication is understood in this context as more frequent and more direct, despite the fact that it occurs via the internet and not face-to-face, so it is not personal. We can expect that the younger generation of academics and students should be particularly more interested in e-learning and the popularity of it should be also investigated from the perspective of cultural changes. The link between better communication and better education is visible, although it is more demanding from the perspective of teachers (more time consuming). The results regarding the D1 variable are unexpected and show that academics undertaking e-learning techniques are dedicated to education and are not avoiding the effort to deliver a high quality of teaching.

In the study, key aspects of the e-learning process, which lower the chances of the decision to use e-learning were shown. The main impediment is connected to the perceived technical problems of preparing and delivering e-learning to students. This aspect will be especially important for academics, who are not familiar with computer and internet technology. The second obstacle which was revealed by our study is that e-learning is perceived as the excessive mechanization of the education process and for many academics, it is not acceptable. In our opinion, there is indeed a threat of excessive mechanization of the education process, and in our opinion, the best method of teaching accounting subjects is to combine traditional and e-learning techniques.

## 5. Conclusions and Further Research

Recent years have brought major changes in higher education. One of the most significant indicators of those changes is the use of e-learning by the most prestigious universities in the US. The Open CourseWare initiative started in 2001, is a free and open publication of material from thousands of MIT courses, covering the entire MIT curriculum and used by millions of learners and educators around the world. The site offers materials from 2340 courses and has 200 million visitors. Another one is Coursera, which was founded in 2012 by two Stanford Computer Science professors, who wanted to share their knowledge and skills with the world. The platform offers complete on-line lectures from the world's top universities and education providers. At the moment, it comprises 1600 courses and has 22 million learners. Although the Polish setting is far distant from the US environment, e-learning has been developing in Poland for several years. Many universities have just started introducing e-learning on a larger scale, but there is still much to be done in this regard. Challenges are faced by all actors involved in this process: educators, universities, and students. This research focuses on the first group of participants—educators.

The aim of the study was to investigate the accounting educators' opinions on the benefits and limitations associated with the e-courses and to find out their views on how e-courses differ from traditional classes. A questionnaire was distributed among the accounting lecturers employed by the leading Polish Economic Universities. Out of 79 respondents, only 19 had experience with e-learning courses. Enhancement of the efficiency of the teaching process was perceived by the Polish lecturers as the main benefit, while a large amount of work associated with designing and updating course materials was regarded as the main difficulty associated with the e-courses. A more thorough comparison of the respondents' answers regarding the use of traditional and distance learning allows the following conclusion to be drawn: Although respondents are familiar with the benefits of e-learning, the time and effort which needs to be put into preparation and delivery of e-courses, as well as the risk of students' dishonest behaviour, seem to be major obstacles to use it as a teaching method on a larger scale.

The performed statistical analysis (MANOVA) supports the first hypothesis, stating that the perception of the effectiveness of e-learning techniques differs between the group of academics, who already use it and the group, which have not decided to use it. The analysis also provides evidence for the second hypothesis, conjecturing that the effectiveness of e-learning, in comparison with traditional teaching, depends on the perceived advantages and disadvantages of e-learning. The results of the study suggest that the factors which influence the effectiveness of e-learning in teaching accounting include: easiness of e-learning delivery, more regular learning process, a greater degree of development of students' social competences during e-learning classes and the more effective process of verification of students' progress in comparison with traditional classes.

The study also investigates the reasons behind the decision to move from the traditional approach to e-learning in teaching accounting subjects. The results suggest that lecturers who decided to use e-learning recognize this way of didactics as more efficient and, at the same time, more demanding in comparison to traditional classes. The third hypothesis states that the most important factor influencing the decision to use e-learning is the better efficiency of the teaching process. Based on the logit regression analysis, the most important aspects of the e-learning process behind the decision to use e-learning are more efficient communication and, unexpectedly, the necessity to invest a large amount of work within designing and updating course materials. The study also reveals the main obstacles which discourage academics from e-learning: technical problems during course preparation and course delivery and a sense of excessive mechanization of the learning process. The results provide evidence supporting the second hypothesis.

Like in every study, there are a few limitations that need to be considered. The first limitation relates to the size of this research sample. With 79 usable respondents, this is rather limited in size. However, this number does not seem to be very low, considering that the respondents were accounting lecturers employed by the leading Polish higher education institution. Secondly, the survey was conducted among Polish educators, which makes it impossible to draw any conclusions on a larger,

international scale. Thirdly, in our analysis, we are comparing the perception of two groups of lecturers; the one, who already engaged in e-learning techniques and has personal experience with regard to the advantages and disadvantages encountered in the education process, and the second group, which has not decided yet for different reasons to use e-learning techniques, with no personal experience. Therefore, the results for these two groups may have different meanings and should be interpreted accordingly. The aim of our study was also to find out how personal experience influences the perception of the advantages and disadvantages of e-learning. Finally, we wanted to reveal the misperception about disadvantages and difficulties, which are shared by academics, who so far haven't decided to use e-learning techniques and have no experience in this matter.

Despite the above limitations, the findings of the research have important implications for policymakers and practitioners. It seems that a stronger commitment to e-learning from the institutions, e.g., with e-learning being fully integrated into the students' curriculum, could encourage educators to implement this method. It appears that guidelines on a national level could also support the development of this form of teaching. The research clearly shows that educators need to develop and improve their skills regarding course design, development and delivery of e-learning, and IT. They should feel supported on each and every step of this process by the university's technical and administrative staff.

It is believed that the presented research could help one to better understand the rationale behind the educators' decision to use e-learning while delivering accounting classes, and the factors that influence the effectiveness of this process. Given the growing interest in e-learning from both lecturers and students, as well as the dynamic development of e-learning tools and techniques, also in Poland, there arise ample opportunities to develop further research in this field. Additional and more detailed analyses could shed more light on e-learning as an educational tool, from both the educators' and students' perspectives. It seems that the approach of higher education institutions to this issue, including their motives, benefits, and difficulties, also poses an interesting research problem. A comparison of the e-learning use in the accounting field between various countries could provide some insights into institutional and cultural factors, influencing the decision to move towards the e-learning approach.

**Author Contributions:** K.G., M.K. and J.K. contributed to the study conception and design. The literature review was performed by M.K. and A.H., data collection was performed by J.K., M.K. and K.G., and statistical analysis by K.G. The first draft of the manuscript was written by M.K. and J.K. and review and editing were performed by A.H. All authors have read and agreed to the published version of the manuscript.

**Funding:** This research was funded by the Ministry of Science and Higher Education, Poland, (financed from the subsidy granted to the Cracow University of Economics).

**Conflicts of Interest:** The authors declare no conflict of interest.

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
