# Peer review of "Embedding E-Learning in Accounting Modules: The Educators’ Perspective"

_education, doi:10.3390/educsci10040097_

Round 1

Reviewer 1 Report

Thank you for the opportunity to review this manuscript, “Embedding e-learning in accounting modules. The educator’s perspective”.

The results of the survey have shown that e-learning is not widely used by accounting academics in Poland. The study provides evidence that lecturers, who decided to use e-learning, perceive this way of teaching as more efficient and at the same time more demanding in comparison to traditional classes. The paper contributes to the understanding of the use of e-learning in accounting education and offers findings that might be useful for both policymakers and practitioners.

After closely reviewing this manuscript I find that there are some areas which need to be addressed before the manuscript is ready for publication. The following suggestions are meant to support further development of the manuscript.

LIMITATIONS

If this study wants to provide evidence that lecturers, who decided to use e-learning, perceive this way of teaching as more efficient in comparison to traditional classe a MANOVA or mean differences should have be done. Data were collected from a small sample of participants; As instruments are used in English, it has to be settle the proficiency in English of all participants. Percentage results data is not the most appropriated analysis for the objectives. The inferences made and the discussions of the implications of the findings are consistent with the empirical results reported. The discussion and conclusions are very short compared to the introduction and review of the literature.

STRENGHTS:

The topic of the research is important and the findings of the study have potentially implications for both professors and university policymakers. The research is supported by a well-developed and generally clear review of the literature. The hypotheses examined appear legitimate in light of the literature reviewed.

Author Response

We appreciate your feedback on the earlier version of the manuscript. The comments have been very helpful in our revision work, and we believe we now have a better paper. Below, we provide an overview of what we have done in response to the issues raised.

Reviewer 2 Report

Dear Authors,

I have read with attention and interest your paper that in my opinion is interesting and stimulating from different points of view.

In spite of this, I have some important questions, related in particular to your methodological approach.

Lines 290 and later – 79 respondents to questionnaires are mentioned but only 19 of them implemented e-learning. The other respondents, as you wrote, based their responses on their projections regarding these issues.

In my opinion it isn’t correct to put together these two types of respondents because their responses are very different in their meanings.

It could be more interesting and rigorous a comparison between different visions and ideas of e-learning from lectures who have experimented it in his/her course and those who haven’t done this. In this way, if you consider your research as an exploratory survey, you would contribute to discuss common and/or wrong ideas about e-learning (even if your sample is quite small).

  • Lines 363 and later – You wrote:The content of the survey consists of categorical questions. The responses to the first group of them are on the scale 1 to 9 and responses to the second group may be presented as dummy variables (zero- one). The application of Likert scale has been considered, but finally abandoned. The authors believed that more variability should be introduced to the data in order to detect statistical relations between variables.

I have not understood what are the “first” and the “second” group (of categorical questions?) you mentioned. I suppose that they are listed in Tables 5 and 9 but it’s not clear why are they divided in two groups?

It’s the first time you mentioned “scale 1 to 9”; what is it?

You wrote: “the application of Likert scale has been considered but finally abandoned”. Why? What does it mean?

To a better comprehension of your questionnaire, in my opinion, could be useful a detailed explanation of this tool and its inclusion as an Appendix.

Lines 379 and later

I have not clear your hypotheses, that seem to me related two different sub-groups of your sample.

H1 seems related to lectures that have not used e-learning methods

H2 seems related to lectures that have decided to use e-learning methods.

Why have you chosen them?

Lines 486 -494

This part seems coherent with the introduction and not with the conclusion section.

I’m sure that if you will give a detailed explanation of the above points, your paper will become more clear and stimulating.

Many thanks for your attention.

My Best Regards

Author Response

(The authors gave the same response as above.)

Reviewer 3 Report

See apended file.

Author Response

(The authors gave the same response as above.)

Round 2

Reviewer 2 Report

Dear Authors, here in attachment my comments to your 2nd version of the manuscript.

My best regards

Author Response

Thank you for the time and effort you invested in reviewing our paper. We appreciate your comments and suggestions. Below, we provide an overview of what we have done in response to the issues raised.

Reviewer 3 Report

Authors improved the text. In the future, the statistical (econometric) expertise would be helpful. When designing an experiment, data analysis, and multivariate statistical inference along with econometric modelling an appropriate expert should be included.

Author Response

(The authors gave the same response as above.)
